# Child stature, maternal education, and early childhood development in Nigeria

**Emmanuel Skoufias** [1]*, **Katja Vinha** [2]

1 The World Bank, Washington, District of Columbia, United States of America, 2 Consultant, Nashville, Tennessee, United States of America

☯ These authors contributed equally to this work.

* eskoufias@worldbank.org

**Data Availability Statement:** All data for the present study were taken from publicly available data sources. In particular, the 2016-17 Nigeria MICS data files are available (upon registration)

## Abstract

Data from the 2016–17 Multiple Indicator Cluster Survey from Nigeria are used to study the relationship between child stature, mother's years of education, and indicators of early childhood development (ECD). The relationships are contrasted between two empirical approaches: the conventional approach whereby control variables are selected in an ad-hoc manner, and the double machine-learning (DML) approach that employs data-driven methods to select controls from a much wider set of variables and thus reducing potential omitted variable bias. Overall, the analysis confirms that maternal education and the incidence of chronic malnutrition have a significant direct effect on measures of early childhood development. The point estimates based on the ad-hoc specification tend to be larger in absolute value than those based on the DML specification. Frequently, the point estimates based on the ad-hoc specification fall inside the confidence interval of the DML point estimates, suggesting that in these cases the omitted variable bias is not serious enough to prevent making causal inferences based on the ad-hoc specification. However, there are instances where the omitted variable bias is sufficiently large for the ad hoc specification to yield a statistically significant relationship when in fact the more robust DML specification suggests there is none. The DML approach also reveals a more complex picture that highlights the role of context. In rural areas, mother's education affects early childhood development both directly and indirectly through its impact on the nutritional status of both older and younger children. In contrast, in urban areas, where the average level of maternal education is much higher, increases in a mother's education have only a direct effect on child ECD measures but no indirect effect through child nutrition. Thus, DML provides a practical and feasible approach to reducing threats to internal validity for robust inferences and policy design based on observational data.

## Introduction

In an ideal world, recommendations for social policies aimed at improving welfare and especially maternal and child outcomes would be based on well-established evidence on the causal relation between the variables of interest. In reality, however, this is not feasible. The

from UNICEF's online database at http://mics.unicef.org/surveys.

**Funding:** The author(s) received no specific funding for this work.

**Competing interests:** The authors have declared that no competing interests exist.

increasing number of randomized control trials (RCTs) useful for establishing causal inferences, combined with the limited ability of RCTs to address every policy relevant question, has not been sufficient to satisfy the increasing demand for evidence-based policy advice. At the same time the higher standards for the internal and external validity of the empirical evidence generated, has increased the urgency to extract credible causal relations, rather than just partial correlations, between variables of interest based on the already available and accessible cross-sectional surveys.

Yet, studies with credible causal inferences based on regression analysis using non-experimental household cross-sectional data are not common, or are even purposefully avoided by young researchers in social sciences aspiring to establish their publication record in their respective fields, in the context of a proliferation of RCTs as the standard for causal inferences. Instead, the common practice in fields such as maternal and child health, is to present estimates from simple regression models with few chosen controls for confounding factors acknowledging the potential role of other confounding factors left out of the model and publish with the qualifying statement that such estimates establish significant partial correlations but not causation between the variables of interest [1]. Frequently the acknowledged relative weaknesses in the internal validity of these studies, is compensated by efforts to strengthen the external validity of the findings by carrying out the same regression (or partial correlation) analysis across different countries with the same type of survey (e.g., [2–4]).

This paper applies recently developed statistical methods to address threats to internal validity in simple regression models. The paper uses individual child data from Nigeria for the purpose of deriving "causal" inferences about the effect of child stature and mother's level of education on early child development (ECD). Short child stature can indicate impaired growth and development that children experience from chronic malnutrition. The level of education of a mother, affects early childhood development through a variety of channels: improved child health and nutrition, increased labor force participation, and income earning opportunities (income effects), greater empowerment and bargaining power within the household, complementarities with household characteristics and community services, access to information (i.e., exposure to media) and information processing [5]. ECD is defined to include the literacy/numeracy, as well as physical, social, emotional, and learning/cognitive development of the child. Parental investments in the quantity and quality of ECD form both the basis for readiness as well as for performance and achievement in education leading to higher productivity. Thus, they are critical for the quantity and quality of the human capital of future generations [6].

Chronic malnutrition is an important factor associated with early childhood development. Impaired growth in early life—particularly in the first 1000 days from conception until the age of two—has adverse functional consequences on the child as these years contain the most rapid changes in brain development. A frequently used measure of chronic malnutrition is stunting which identifies short stature children based on their height (length) in comparison to the height (length) of well-nourished children of the same age and sex. Stunting is associated with concurrent and later cognitive delay or deficit [7–9], poor cognition [10–12] and educational performance [13], low earnings in adulthood [14], lost productivity and, when accompanied by excessive weight gain later in childhood, an increased risk of nutrition-related chronic diseases as adults [15].

Given the difficulties involved in the measurement of developmental delays and some empirical evidence on the causal effect of stunting on early childhood development, stunting, which is more easily measured, is frequently used as a stand-in for developmental delay in cross-sectional population health [3, 4]. However, there is an ongoing debate about the relationship between child linear growth (height gain) and child development, especially based on

the evidence from cross-sectional data. On the one hand, some researchers argue that there is a causal relationship between linear growth or stunting and childhood development, and especially their cognitive ability [16, 17]. On the other hand, studies such as Tran et al. based on data from 178,393 children aged 36 to 59 months from 55 countries conclude that the "association between growth and development in early childhood appears to be primarily a co-occurrence because the magnitude of the association varies among settings from no association in higher-income countries to a moderate level in low-income countries, and that. . ..overall, growth is not a sensitive and therefore suitable indicator of child development" [18].

There is also plenty of empirical evidence available in the literature on the association between a mother's education and different dimensions of child health such as infant mortality, the number of immunizations [19], child nutrition measured by height-for-age (HAZ) or stunting [20], and fertility [21–24]. Yet, causal evidence on the effect of a mother's education on ECD net of its effect of stunting is rather scarce. A better understanding of the channels through which maternal education affects ECD outcomes is essential for the prioritization of policies related to child health. Evidence of a direct effect of maternal education on ECD, separate from its potential effects on child stunting, would suggest that maternal education has the advantage to compensate for the potentially negative effects of stunting on ECD and thus provide even stronger justification of the need to increase the quantity and quality of female education as an especially important factor in the development and economic growth of Sub-Saharan Africa countries undergoing a demographic transition.

With this background in mind, the analysis in this paper takes advantage of the variation across households in indicators of early childhood development and applies recently developed methods for more robust inferences based on observational data [25]. The results from the conventional ad-hoc specification of a model, i.e., a regression model with the usual suspect confounding factors, chosen based either on custom or economic intuition, are contrasted to a more sophisticated approach, called double machine learning (DML). DML employs data-driven methods to select controls from a much wider set of variables available in the survey along with statistical learning methods that yield information that would not be available from fitting the model only once using the original sample [26–28]. For a parallel analysis using MICS data from the Republic of Congo and Sao Tome & Principe, yielding similar patterns in the differences between ad-hoc and DML estimates see [29]. Some of the methods employed here have also been applied recently in predicting the prevalence of low cognitive and/or socio-emotional early childhood development scores but not in exploring the causal relationship of mother's education and child nutrition with ECD outcomes [30].

## Data

The analysis is based on the cross-sectional Multiple Indicators Cluster Survey (MICS) for Nigeria collected between September 2016 and January 2017. The MICS is an international household survey program that collects information about health, nutrition, education, and early development of well-being of children and their families in developing countries. The survey covers 10,773 children between 36 months and 59 months of age and 69 percent of these children live in rural areas. There were a total of 28,578 children under five with a response rate of 98.3% [31] The majority of the children are in the poorer northern zones, (17 percent in North-Central, 22 percent in North-East, and 38 percent in North-West). Only twenty three percent of the children are in the Southern zones (5 percent in South-East, 8 percent in South-South and 10 percent in South-West).

In statistical terms, a child is stunted if his/her height-for-age Z-score (HAZ) is more than 2 standard deviations (SD) below the median height of a healthy reference population (that is,

HAZ <-2). The measure of HAZ and thus stunting used in this study is based on the new WHO child growth reference population. A child experiencing stunting is considered stunted. The incidence of stunting among children ages 36–59 months is a consequence of the growth faltering in the first 1,000 days from conception. Growth faltering is the rapid decline in height- and weight-for-age of children in the first two years of life and is common in many developing countries. Growth faltering among children was first documented in a study by Shrimpton et al [32] and to a large extent, it is in response to these findings that several global health policy and information campaigns with emphasis on the first 1,000 days window have been initiated [33].

Figs 1 and 2 present the cross-sectional age profile of growth faltering (HAZ scores) for children between 0 and 59 months of age at the national level and for urban and rural areas separately. As is easily apparent, growth faltering in the first 1,000 days is prevalent among children in Nigeria, and especially in the rural areas, with growth faltering (or HAZ decreasing) accelerating rapidly with age in the first three years of life, somewhat recovering after 35 months of age. For a recent cross-country study on the determinants of growth faltering, see [34]. The growth faltering curves are derived using a local polynomial smooth of HAZ with the Stata command "lpoly".

Much of the empirical literature on the role of a mother's education focuses on children less than 60 months without making any distinction about the age of the child; exceptions are [35, 36]. Given the rapid growth faltering in the first two years of a child's life and the fact that measures of ECD are available only for children 36–59 months old, this study investigates not only whether there is a causal relationship in the full sample of children 0 to 59 months of age, but also whether there are any structural differences in the causal relation between maternal education and child stature separately in the sample of younger (0–35 months) and older (36–59 months) children.

To construct the outcomes of interest, information is used from the child development module which collects information on children aged 36 to 59 months. The survey asks for information on 10 competences in four child development domains: literacy/numeracy (3 questions), physical (2 questions), socio-emotional (2 questions), and learning (3 questions). Based on these questions six different measures are constructed. The 10 items were determined through multi-country field tests, validity and reliability studies, and consultations with experts. For more details on the UNICEF method for constructing the ECDI see Loizillon et al. [37]. The first measure is the MICS-specific measure developed by UNICEF whereby a child is developmentally on track if he/she is on track in at least three of the four domains [37]. The definitions for domain competences are given in Table 1. In this paper, an alternative composite score for early child development, the standardized Early Child Developmental Index (or ECDI), is also employed. ECDI is the total number of positive responses to the 10 questions without considering the domain of the question. This score is normalized to a z-score with mean of 0 and standard deviation of 1. This measure is a more sensitive, though less scientifically based, measure of child development as it simply compares the developmental status of any given child to the developmental status of average child in the sample. Furthermore, competency at each of the four domains are considered as separate ECD outcomes of interest.

For the analysis to be meaningful variation in the outcomes is necessary. The only domain with near universal competence is physical development (Table 1). Given the information collected, this domain captures only significant delays in physical development since pincer grasp, the first component, is usually achieved by 12 months of age and this competency is applied to the sample of children 36 months or older. The other component, that a child is too sick to play, is a subjective assessment by the mother and may reflect recent behaviors and illnesses rather than developmental delays.

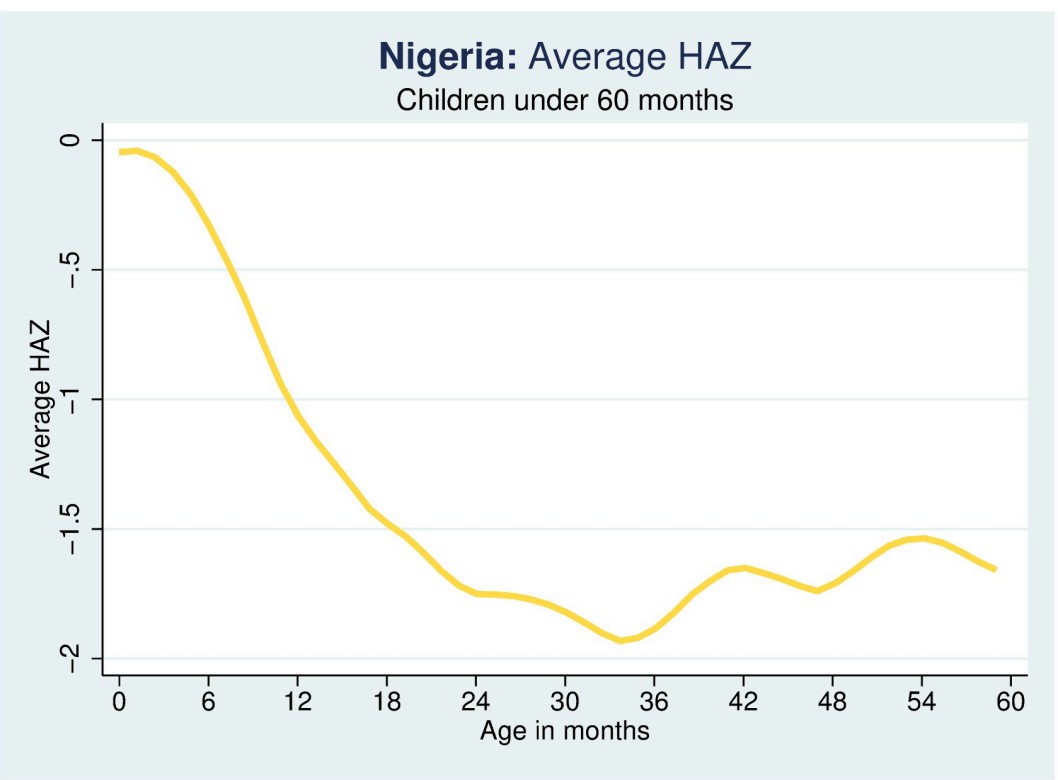

**Fig 1. Growth faltering at the national level.** Source: Author calculations using Nigeria MICS 2016–17.

The key variables of interest are in Table 1. See S1 Table for the summary statistics of some of the control variables used in the analysis. What is evident is that in terms of ECD outcomes there are urban-rural differences as well as heterogeneity across zones. Urban children perform, on average, 13 percentage points higher on the total ECD score than rural children. Furthermore, the share of children on track in each of the 4 domains is higher in urban areas than in rural areas.

Children in the three better off southern zones are more likely to meet the criteria for most of the domains. In the southern zones 86 percent, 85 percent, and 79 percent of children are on track in at least 3 of the 4 domains in South-West, South-East and South-South zones, respectively. In the relatively poorer northern zones fewer children are on track with 70 percent, 53 percent, and 52 percent of children on track in North-Central, North-East, and North-West, respectively.

The domain with the most distinct difference is literacy and numeracy. In the southern zones at least two-thirds of the children are on track whereas in the northern zones less than one-third are on track. Specifically, in North-East only 13 percent of the children are on track whereas in South-East 71 percent are on track. Both in North-East and in North-West a smaller share of children are on track in literacy and numeracy than in rural areas in general. The domain with no sharp difference between northern and southern zones is socio-emotional development: in South-South 69 percent of the children are making progress in the domain similar to North-East and North-West rates, 70 percent and 67 percent, but below the 78 percent in the North-Central zone.

There are differences in the nutritional outcomes between urban and rural children as well as across the geopolitical zones. The average height-for-age in rural areas is 2.11 standard

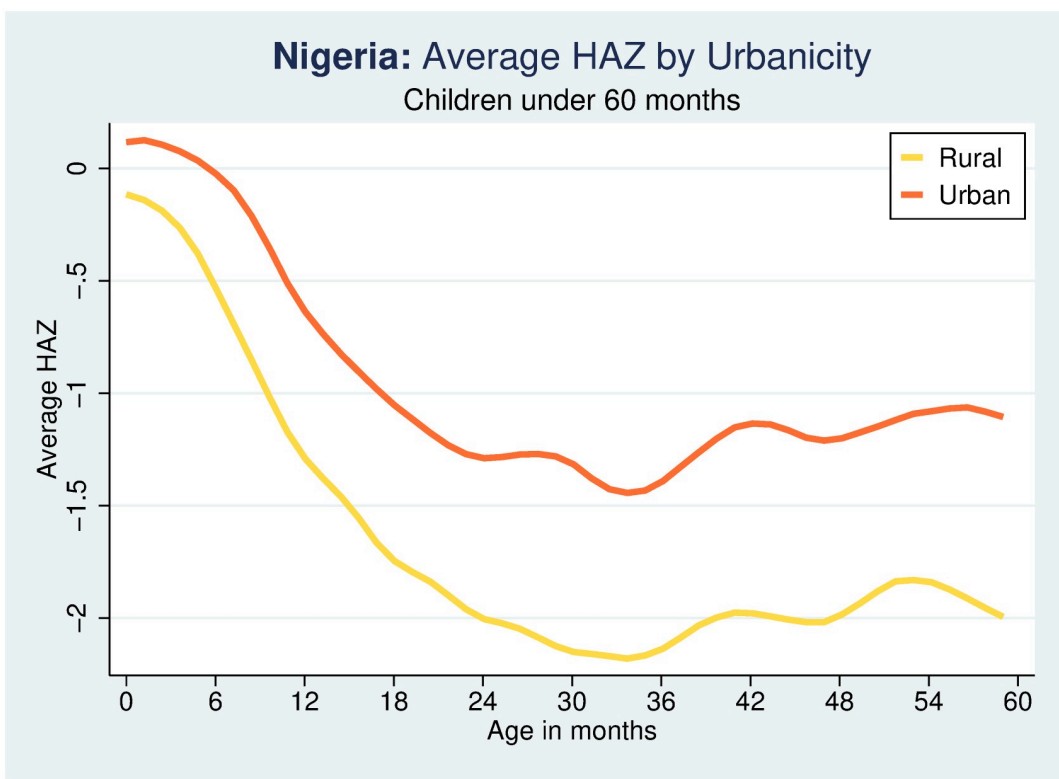

**Fig 2. Growth faltering in urban and rural areas.** Source: Author calculations using Nigeria MICS 2016–17.

deviations below the median height-for-age of healthy children and 53 percent of children are stunted. In urban areas the average-height-for-age is negative but at -1.34 above the -2 threshold for stunting and "only" 30 percent of children are stunted. Similarly, for underweight a larger share of rural children (31 percent) are underweight than of urban children (20 percent). In terms of the geo-political zones, northern zones have much higher undernutrition rates than southern zones. North-West has the highest rates of undernutrition and the lowest rates are found in South-East.

Mother's education follows the above trends as well. Urban mothers tend to be more educated, with mean years of education at 8.32 years in comparison with rural mothers who on average have 3.27 years of education. The differences across zones are large. Mothers in the North-Central zone have about half the years of education than mothers in the southern zones, and mothers in the North-East and North-West zones have less than one-third the years of education than mothers in the South.

## Methodology

The main question addressed in this paper is an inference question: how does the predicted value of an indicator of early childhood development change, if the incidence of stunting (a binary variable), or mother's years of education are increased by one unit, holding other confounding factors fixed? This question can be answered within the context of a partially linear model, such as

$$Y = \beta_1 D_1 + \beta_2 D_2 + f(Z) + \epsilon \tag{1}$$

**Table 1. Summary statistics for early childhood development outcomes for children 36–59 months.**

| | Full Sample | Urban | Rural | North-Central | North-East | North-West | South-East | South-South | South-West |
|---|---|---|---|---|---|---|---|---|---|
| **Child height-for-age (HAZ)** | -1.87 | -1.34 | -2.11 | -1.56 | -2.21 | -2.45 | -0.60 | -0.81 | -0.98 |
| **Child is stunted** (= 1 if HAZ<= -2, 0 otherwise) | 0.46 | 0.30 | 0.53 | 0.35 | 0.56 | 0.63 | 0.13 | 0.18 | 0.18 |
| **Child is underweight** (= 1 if WAZ<= -2, 0 otherwise) | 0.28 | 0.20 | 0.31 | 0.15 | 0.38 | 0.38 | 0.08 | 0.11 | 0.13 |
| **Child is wasted** (= 1 if WHZ<= -2, 0 otherwise) | 0.05 | 0.06 | 0.05 | 0.03 | 0.06 | 0.06 | 0.04 | 0.03 | 0.05 |
| **Years of education of the mother** | 4.83 | 8.32 | 3.27 | 5.20 | 2.96 | 2.58 | 10.33 | 9.77 | 10.02 |
| **On track in ECD (UNICEF):** = 1 if child is on track in at least 3 of the 4 domains, = 0 otherwise | 0.63 | 0.76 | 0.57 | 0.70 | 0.53 | 0.52 | 0.85 | 0.79 | 0.86 |
| **Total score (ranging from 0 to 10) in 10 ECD questions (prior to constructing z-scores):** number of positive responses across the literacy/numeracy, socioemotional, learning, and physical domain items | 5.53 | 6.42 | 5.14 | 5.65 | 4.81 | 4.97 | 6.98 | 6.96 | 7.17 |
| **On track in literacy/numeracy:** = 1 if the child can do **at least two of the following**: identify/name at least ten letters of the alphabet; read at least four simple, popular words; and/or know the name and recognize the symbols of all numbers from 1 to 10 | 0.30 | 0.51 | 0.20 | 0.29 | 0.13 | 0.15 | 0.71 | 0.67 | 0.70 |
| **On track in physical development:** = 1 If the child can pick up a small object with two fingers, like a stick or rock from the ground, and/or the mother/caregiver does <u>not</u> indicate that the child is sometimes too sick to play | 0.92 | 0.95 | 0.91 | 0.93 | 0.88 | 0.92 | 0.95 | 0.94 | 0.94 |
| **On track in learning** = 1 if on track **if one or both of the following are true**: child follows simple directions on how to do something correctly; when given something to do, is able to do it independently | 0.79 | 0.86 | 0.76 | 0.79 | 0.73 | 0.76 | 0.84 | 0.90 | 0.90 |
| **On track in socio-emotional development:** = 1 if two of the following are true: the child gets along well with other children; the child does <u>not</u> kick, bite or hit other children; and the child does <u>not</u> get distracted easily | 0.72 | 0.74 | 0.71 | 0.78 | 0.70 | 0.67 | 0.82 | 0.69 | 0.83 |

Source: Author calculations using Nigeria MICS 2016–17.

where $Y$ denotes the ECD outcome, $D_1$ is the HAZ score of the child (or a binary variable identifying whether the child is stunted), $D_2$ is the years of education of the mother of the child, and $f(Z)$ is a possibly nonlinear function of the confounding factors or controls $Z$. In the discussion following it is assumed that inferences are desired on the two variables, $D_1$, $D_2$, whose respective coefficients $\beta_1$, and $\beta_2$ provide an approximation of the long-run relationship between the variable and the outcome of interest.

Traditionally, in the empirical literature in social sciences, the number of control variables in $Z$ is determined a priori, or in an ad-hoc manner, rather than choosing the control variables in a data-dependent manner. For example, in the health and nutrition literature (e.g., [3, 4, 20]) the conventional approach consists of selecting control variables based on common sense and/or the economic intuition [33] derived from the human capital and health production function model [38].

The paper presents results following this ad-hoc approach, by assuming a small set of control variables entering Eq (1) linearly, i.e.,

$$Y = \beta_1 D_1 + \beta_2 D_2 + \varphi X + \epsilon \tag{1a}$$

In this setting, the coefficients $\beta_1$, and $\beta_2$ provide estimates of the partial correlation between $D_1$, and $D_2$, with the dependent variable net of the effect of all other confounding factors. The control variables in Eq (1a) are denoted by $X$ instead of $Z$ to make explicit the point

that this is a subset of selected control variables from the wider set of control variables available in the survey and denoted in Eq (1) above by $Z$. The 23 control variables used in the regressions, include a binary variable for girls, child age (in months) and age squared, household size, the dependency ratios, four binary variables identifying the household's wealth quintile, a binary variable for rural areas and a set of binary variables for region of residence. The wealth index is composed of: type of floor, roof, wall, fuel used by household for cooking, household assets, source and location of drinking water and sanitation facility [38, p. 14].

It is well established in the literature that the coefficients estimated from a regression have a causal interpretation if the conditional mean independence assumption holds (e.g., [39]). The conditional independence assumption (CIA), in essence, requires that $D_1$ and $D_2$ are conditionally exogenous, which implies that that $D_1$ and $D_2$ are as good as randomly assigned conditional on the covariates $X$. The CIA for internal validity is weaker than the *conditional mean zero assumption* of $E[\epsilon \mid X, D_1, D_2] = 0$ typically required for the OLS method. As long as $D_1$ and $D_2$, conditional on a set of control variables, happen to be independent of the error term, causal inferences can be made about $D_1$ and $D_2$, even if some or all of the control variables were correlated with the error term. Given that only a few controls are included in $X$, there is a variety of threats to the internal validity of the causal interpretation of the estimated coefficients $\beta_1$, and $\beta_2$, arising from the potential of correlation between $D_1$ and $D_2$ and the error term $\epsilon$. Omitted variable bias, for example, can lead to a nonzero correlation between the error term and $D_1$ and $D_2$ leading to a violation of the conditional independence assumption (as well as of the conditional mean-zero-assumption, i.e., $E[\epsilon \mid X, D_1, D_2] = 0$). It is precisely for this reason, that the estimates of the coefficients $\beta_1$, and $\beta_2$ from a regression model as (1) based on the ad-hoc selection of a few controls, are typically interpreted as estimates of the association or partial correlations of $Y$, $D_1$, and $D_2$, rather than as causal estimates of the effect of $D_1$, and $D_2$, on $Y$.

To address the preceding concerns, the "double machine-learning" (henceforth DML) approach developed by Chernozhukov et al. for the purpose of making inferences is adopted [25]. Expanding the set of controls to a much larger set of variables, denoted by $Z$, minimizes potential omitted variable bias, and this in turn makes the conditional independence assumption (CIA) required for causal inferences much more defensible than in the case of a few controls $X$ selected on an hoc basis. As implied by the name, DML allows for a data-driven selection of controls from a wider set of variables $Z$ based upon the combination of methods of statistical learning, such as regularization, resampling ([26], [27]) and cross-fit partialling-out [25] Regularization (or shrinkage) can fit a model containing a large number of regressors, even greater than the number of observations available, using a technique that constrains or regularizes the coefficient estimates, or equivalently, that shrinks the coefficient estimates towards zero (e.g., LASSO) [40]. Resampling methods that repeatedly draw samples from a training set and refit the model of interest on each auxiliary (test) sample, combined with cross-fit partialling-out that swaps the roles of training and auxiliary samples to obtain multiple estimates can provide additional information about the bias and efficiency of the fitted model. Such cross-validation sampling methods, yield information that would not be available from fitting the model only once using the original training sample. In combination, the data-driven selection of control variables for the model from a much larger pool of variables combined with resampling and cross-fit partialling-out offer the opportunity to increase robustness, reduce the variance of the coefficient estimates and to avoid overfitting (i.e., poor predictions on future observations not used in model training).

Specifically, the full partially linear model, is specified as

$$Y = \beta_1 D_1 + \beta_2 D_2 + f(Z) + \epsilon \qquad E[\epsilon | Z, D_1, \ D_2] = 0 \tag{2}$$

$$D_1 = g(Z) + v_1 \qquad E[v_1|Z] = 0 \qquad\qquad (2a)$$

$$D_2 = h(Z) + v_2 \qquad E[v_2|Z] = 0 \qquad\qquad (2b)$$

where $Y$, $D_1$, $D_2$ are as above, $f(Z)$, and $g(Z)$ are possibly nonlinear functions of the controls $Z$ summarized by a "high-dimensional" vector of confounding factors that could include many, if not all, of the variables in the survey (and perhaps even greater than the number of observations in the survey).

Eq (2) is the main equation, and $\beta_1$ and $\beta_2$ are the main regression coefficients for inference. If $D_1$ and $D_2$ are exogenous conditional on controls $Z$, $\beta_1$ and $\beta_2$, have the interpretation of the treatment effect parameters. The second and third Eqs (2a) and (2b) keep track of confounding, namely the dependence of the treatment variables on controls, with the functions $g$ and $h$ allowing for the possibility that the functional form summarizing the relationship between the treatment variables $D_1$ and $D_2$ and $Z$ may differ. Eqs (2a) and (2b) are not of interest per se, but they are important for characterizing and removing regularization bias.

As pointed out by Chernozhukov et al. [25], one naive approach trying to improve on the ad-hoc-specification, would likely ignore Eqs (2a) and (2b) and focus only on the first Eq (2) by attempting to select control variables from a much wider pool of variables available in a survey, by applying LASSO to Eq (2) above, forcing the treatment variables $D_1$ and $D_2$ to remain in the model by excluding $\beta_1$ and $\beta_2$ from the LASSO penalty. One could then try to estimate and do inference about $\beta_1$ and $\beta_2$ by applying ordinary least squares with $Y$ as the outcome, and $D_1$ and $D_2$ and any selected control variables as regressors. The problem with this approach can be seen by noting that, in LASSO, any variable that is highly correlated to the $D_1$ or $D_2$ treatment variables is likely to be dropped since including such a variable does not add much predictive power for the outcome $Y$ given that the treatment variables $D_1$ and $D_2$ are already in the model. As a consequence, the exclusion of a variable that is highly correlated to the treatment variables $D_1$ and $D_2$ can lead to substantial omitted-variables bias in the coefficients $\beta_1$ and $\beta_2$. Similarly, if one applied a variable selection method to only Eqs (2a) and (2b) for predicting $D_1$ and $D_2$, one could miss variables that have moderate-sized coefficients in predicting $D_1$ and $D_2$, but large direct effects on $Y$. Such an omission may again lead to non-negligible omitted-variable bias in the coefficients $\beta_1$ and $\beta_2$. The DML approach has a more relaxed sparsity requirement than double selection [41], that adds robustness since the sample is split and coefficients are obtained from one sample and used in another independent sample (see S1 Appendix for more details).

In more intuitive terms, the DML method consists of estimating the coefficients $\beta_1$ and $\beta_2$ from the regression coefficients of the "residualized" $Y$ on the "residualized" $D_1$ and $D_2$ variables, where the residuals are constructed from the difference between the actual value of $Y$ (and the actual values of $D_1$ and $D_2$) and the fitted value from a regression of $Y$ (and $D_1$ and $D_2$) on $Z$. The sequence of steps involved in applying the DML method is presented in detail in S1 Appendix. The maintained (and untested) assumption for causal inferences based on the DML regression Eqs (2), (2a) and (2b) is that $E[\epsilon \mid Z, D_1, D_2] = 0$, and $E[v_i \mid Z] = 0$ for $i = 1, 2$. An alternative approach is to employ instrumental variables for the $D_1$ and $D_2$ treatment variables and apply a variant of the DML approach explored here. The difficulty common to most applications of the instrumental variable approach is to identify variables that are excluded from Eq (2) but correlated with the treatment variables $D_1$ and $D_2$ and uncorrelated with the error term $\epsilon$. The key difference between the ad-hoc conventional approach and DML that allows for causal inferences is the assumption that $D_1$ or $D_2$ are conditionally exogenous (or as

good as randomly assigned conditional on the covariates $Z$), is more defensible. The vector of the confounding variables $Z$ is of much higher dimension than the vector $X$ which is composed of the few confounding variables chosen in ad-hoc manner. Thus, in the ad-hoc approach the estimates of $\beta_1$ and $\beta_2$ may subject to omitted variable bias thus preventing one from making causal inferences.

In contrast to the ad-hoc approach, the vector $Z$ in the DML approach as applied in this study, consists of 451 possible control variables for the analysis. The variables in the control vector $Z$ are also subject to some pre-selection as in the ad-hoc specification, based on the criterion that these variables are not likely to be correlated with the error term (i.e., they are exogenous) to ensure that the conditional mean-zero assumption is more likely to hold $E[\epsilon \mid Z, D_1, D_2] = 0$. Based on this criterion, variables such as whether the child is attending an early childhood development program and variables measuring the quality of care, such as the availability of playthings, or books, or the child being left alone in the care of another child, all available in the MICS, are excluded from the vector of controls. These variables are likely to be determined at the same time as early childhood development as well as child undernutrition (stunting), and thus likely to be correlated with the error term if they were included in the set of controls. This also supported by Loizillon et al. [37] who in their report of the development of the ECD index state that "having books and participating in early learning programs were the strongest predictors of children's outcomes as measured by the ECDI. Also, children with more books, playthings from more sources, and left alone less had higher mean item values on the ECDI than children with fewer books, playthings from fewer sources, and left alone more often. The differences were statistically significant."

The list of potential control variables includes, the young and old dependency ratio (ratio of household members younger than 15 years old or older than 64 and the total number of working age members 15–64 years old), child gender, child age (in months) and age squared, whether the child is the son/daughter of the household head, the level of education of the head of the household, household size, the mother's age at birth, the birth order of the child, the number of mosquito nets, the ethnicity and religion of the household, whether the mother believes that wife beating is acceptable in some occasions, whether at least once per week the mother reads newspaper, listens to radio, or watches TV, the type of union between the mother and her spouse, expanded set of the components of the household's wealth index included as separate binary or continuous variables, a binary variable for rural areas and a set of binary variables for regions. The expanded wealth index variables included are: number of persons per room used for sleeping, primary material used for floor, roof, and walls, ownership of household collective assets (electricity, radio, TV, fixed telephone, refrigerator, VCR, sewing machine, clock, generator, computer, water heater), ownership of individual assets (watch, portable telephone, bicycle, motorcycle, car, boat with motor, animal-drawn cart, tricycle), ownership of house, bank account, agricultural land, number of specific animals owned (livestock, horses, goats, sheep, chicken, pigs, camels, ducks, geese, quail, cultured fish), principal source of drinking water for the household and its geographic location, and type of toilet used by the household and whether it is shared or not, and the presence of a handwashing station. When the sample used is not limited to urban or rural areas, the preceding variables, except for age and age squared, are also interacted with the binary variable for rural areas to allow for possible differences between urban and rural areas.

The analysis is carried out using the *xporegress* and *xpologit* commands in Stata v16.1. The commands apply the DML (cross-fit partialling out) method for continuous and binary dependent variables, respectively. The selection of the controls in the DML is based on LASSO by splitting the sample randomly into 10 parts (i.e., LASSO is estimated 10 times). In addition, for the nutrition outcome analysis as well as the composite ECD measures the 10-part random

**Table 2. Effect of stunting and mother's education on early childhood development outcomes.**

| Early Childhood Development outcome | Ad hoc | | | | DML | | | |
|---|---|---|---|---|---|---|---|---|
| | Stunting | | Mother's Education | | Stunting | | Mother's Education | |
| | Coeff | p-value | Coeff | p-value | Coeff | p-value | Coeff | p-value |
| ECDI | -0.194 | 0.000 | 0.030 | 0.000 | -0.184 | 0.000 | 0.018 | 0.000 |
| Being on track on: | OR | p-value | OR | p-value | OR | p-value | OR | p-value |
| 3 out of 4 domains (UNICEF ECD) | 0.684 | 0.000 | 1.054 | 0.000 | 0.711 | 0.000 | 1.027 | 0.001 |
| Literacy and numeracy | 0.529 | 0.000 | 1.104 | 0.000 | 0.563 | 0.000 | 1.073 | 0.000 |
| Physical development | 0.702 | 0.000 | 1.049 | 0.000 | 0.734 | 0.001 | 1.046 | 0.001 |
| Learning | 0.670 | 0.000 | 1.033 | 0.000 | 0.709 | 0.000 | 1.010 | 0.297 |
| Socio-emotional | 0.957 | 0.429 | 1.011 | 0.086 | 0.970 | 0.596 | 0.991 | 0.216 |

Source: Author calculations using Nigeria MICS 2016–17.

splitting is repeated 5 times (and then averaged) to take into account the fact that the selection of variables introduces a new source of variability that may affect the standard errors of the estimates and thus inference. Thus, a total of 50 LASSOs are estimated. The variables selected based on LASSO in one sample may differ for the variables selected in another sample, which implies that the standard errors used for proper inferences would need to take that new source of variability into account. For the specific domain analyses the 10-fold random splitting is only repeated once.

## Results and discussion

Table 2 presents the estimates of the relationship between child stunting and maternal education using the ad-hoc approach of selecting controls and the DML approach, for two alternative measures of ECD, the standardized ECD index (ECDI) and the binary variable identifying whether a child is on track in early childhood development (ECD UNICEF). The dichotomous variable identifying whether a child is stunted allows for a direct comparison of the ECD measures between the groups of children with low HAZ scores (HAZ<-2) and higher HAZ scores (HAZ> = -2). Coefficient estimates with p-values less than 0.05 are shaded in grey to facilitate the discussion of the results.

In general, the ad hoc and the DML approach yield similar impacts from the child's nutritional status and mother's education on the various ECD outcomes. However, the size of the impact estimates is generally lower when using DML rather than an ad hoc approach. According to the ad-hoc approach, one additional year of education of the mother is associated with an increase in the early childhood development index by 0.030 standard deviations, and if a child is stunted the ECDI is 0.194 standard deviations lower (see Table 2). In contrast, based on the DML approach, one additional year of education of the mother increases the early childhood development index by 0.018 standard deviations, and if a child is stunted the ECDI is 0.184 standard deviations lower (see Table 2).

Using the alternative measure of early childhood development (ECD UNICEF), the importance of both nutrition and mother's education on ECD holds (Table 2). According to the ad-hoc approach, one additional year of education of the mother implies that the odds of a child is on track in at least 3 of the 4 development domains are 1.054 higher, while the odds of being on track on ECD outcomes change by -32.6%, or (0.684–1)*100%, among stunted children. In contrast, based on the DML approach, one additional year of education of the mother increases the odds of being on track in ECD by 2.7%, while the odds of being on track on ECD outcomes change by -28.9%, or (0.711–1)*100%, among stunted children.

**Table 3. Effects of stunting and mother's education on ECD in rural and urban areas (children 35–59 months old).**

| | Ad hoc | | | | DML | | | |
|---|---|---|---|---|---|---|---|---|
| Early Childhood Development outcome | Stunting | | Mother's Education | | Stunting | | Mother's Education | |
| RURAL | Coeff | p-value | Coeff | p-value | Coeff | p-value | Coeff | p-value |
| ECDI | -0.192 | 0.000 | 0.028 | 0.000 | -0.185 | 0.000 | 0.021 | 0.000 |
| Being on track in: | OR | p-value | OR | p-value | OR | p-value | OR | p-value |
| 3 out of 4 domains (UNICEF ECD) | 0.690 | 0.000 | 1.042 | 0.000 | 0.710 | 0.000 | 1.020 | 0.025 |
| Literacy and numeracy | 0.553 | 0.000 | 1.105 | 0.000 | 0.595 | 0.000 | 1.081 | 0.000 |
| Physical development | 0.680 | 0.000 | 1.034 | 0.015 | 0.728 | 0.002 | 1.028 | 0.043 |
| Learning | 0.684 | 0.000 | 1.021 | 0.034 | 0.723 | 0.000 | 1.000 | 0.992 |
| Socio-emotional | 0.928 | 0.238 | 1.011 | 0.181 | 0.924 | 0.224 | 0.998 | 0.799 |
| URBAN | Coeff | p-value | Coeff | p-value | Coeff | p-value | Coeff | p-value |
| ECDI | -0.197 | 0.000 | 0.033 | 0.000 | -0.175 | 0.000 | 0.019 | 0.000 |
| Being on track in: | OR | p-value | OR | p-value | | | | |
| 3 out of 4 domains (UNICEF ECD) | 0.663 | 0.001 | 1.072 | 0.000 | 0.733 | 0.017 | 1.042 | 0.003 |
| Literacy and numeracy | 0.468 | 0.000 | 1.111 | 0.000 | 0.497 | 0.000 | 1.078 | 0.000 |
| Physical development | 0.767 | 0.199 | 1.073 | 0.001 | 0.846 | 0.537 | 1.073 | 0.009 |
| Learning | 0.619 | 0.001 | 1.056 | 0.000 | 0.631 | 0.001 | 1.020 | 0.101 |
| Socio-emotional | 1.073 | 0.556 | 1.011 | 0.350 | 1.150 | 0.224 | 0.977 | 0.039 |

Source: Author calculations using Nigeria MICS 2016–17.

The same patterns emerge when the impacts of mother's education and child stunting are compared between the ad-hoc and the DML approach for the specific domains of ECD such as a child being on track on literacy and numeracy skills, physical development and learning. The nutritional status of the child and the mother's education appear to have no impact on a child being on track in his/her socio-emotional development. The same general patterns also hold when the impacts of mother's education and child stunting are compared between the ad-hoc and the DML approach applied separately to urban and rural areas separately (see Table 3). Separate DML estimates for each of the 6 zones of Nigeria are presented in S2 Table. While there is some heterogeneity in the estimates among zones, the estimates for the 3 Northern zones that are primarily rural and 3 Southern zones thar more urbanized do not differ substantially from the rural and urban estimates discussed here. One notable difference is that in both rural and urban areas, the DML approach suggests that mother's education has no impact on a child being on track in learning, whereas the ad-hoc approach suggests that it does. This result may reflect the fact that the learning domain consists of more subjective evaluations of child behaviors which is likely to lead to potential correlation of the education of the mother with unobserved variables included in the error term. The data-driven choice of controls based on the DML approach appears to eliminate the potential omitted variable bias in the estimates based on the ad-hoc approach.

The estimates discussed so far shed light on the direct effect of a mother's education on ECD measures net of its potential positive effects on child ECD through improved child stunting. A more complete picture of the impacts of mother education on child ECD can be obtained by estimating the indirect effect that increased maternal education may have on child early childhood development through its effect on the improved nutritional status of the child. To this end, the impact of a mother's education is analyzed using two alternative measures of the nutritional status of the child: (i) child HAZ, a continuous variable, and (ii) the stunting status of a child, a binary variable. Also, in consideration of the differences in the pattern of

Table 4. Effect of mother's education on nutritional outcomes.

| Nutritional outcome | Ad hoc | | | | | | DML | | | | | |
|---|---|---|---|---|---|---|---|---|---|---|---|---|
| | 0–59 month old | | 36–59 month old | | 0–35 month old | | 0–59 month old | | 36–59 month old | | 0–35 month old | |
| | Coeff | P | Coeff | P | Coeff | P | Coeff | p | Coeff | p | Coeff | p |
| HAZ, All | 0.025 | 0.000 | 0.030 | 0.000 | 0.023 | 0.000 | 0.009 | 0.005 | 0.013 | 0.006 | 0.007 | 0.113 |
| HAZ, Rural | 0.021 | 0.000 | 0.023 | 0.000 | 0.019 | 0.000 | 0.009 | 0.019 | 0.014 | 0.015 | 0.007 | 0.200 |
| HAZ, Urban | 0.031 | 0.000 | 0.041 | 0.000 | 0.026 | 0.000 | 0.009 | 0.081 | 0.010 | 0.194 | 0.010 | 0.152 |
| | OR | P | OR | P | OR | P | OR | p | OR | p | OR | p |
| Stunting | 0.964 | 0.000 | 0.961 | 0.000 | 0.965 | 0.000 | 0.983 | 0.000 | 0.979 | 0.013 | 0.989 | 0.104 |
| Stunting, Rural | 0.964 | 0.000 | 0.962 | 0.000 | 0.964 | 0.000 | 0.977 | 0.000 | 0.971 | 0.001 | 0.979 | 0.005 |
| Stunting, Urban | 0.967 | 0.000 | 0.958 | 0.000 | 0.970 | 0.001 | 0.990 | 0.184 | 0.984 | 0.197 | 0.993 | 0.449 |

Source: Author calculations using Nigeria MICS 2016–17.

growth faltering summarized in Fig 1, separate estimates are provided for older (36–59 months old) and younger (0–35 months old) children.

Table 4 confirms that In Nigeria, mother's education has a statistically significant role in the nutrition status of children. The ad hoc approach yields larger estimates and is more likely to yield statistically significant relationships between mother's education and child nutrition outcomes. These differences are most pronounced for the younger cohort of children and for urban children. So, while the more commonly employed ad hoc approach suggests statistically significant effects from mother's education on a child's nutritional status, allowing for a more complete set of covariates these effects become smaller in magnitude and in some cases lose their statistical significance.

At the national level, based on the DML approach an additional year of education of the mother increases the height-for-age z-score by 0.009 standard deviations for children under 5 years of age. Also, this impact appears to be driven by children who are 36 months or older. In contrast, the ad-hoc approach at the national level suggests that (i) the effect of a mother's education on child HAZ is almost 3 times higher (0.025 vs. 0.009); and (ii) mother's education has a statistically significant effect on the HAZ of older as well as younger children (less than 36 months of age). Maternal education does not appear to have a statistically significant effect on the incidence of child wasting, or low weight for height, the measure most affected by acute significant food shortage and/or disease and a strong predictor of mortality among children under five.

The separate estimates based on the DML approach for urban and rural areas, reveal that mother's education matters to a child's chronic malnutrition status measured by whether the child is stunted or not in rural areas but not in urban areas (Table 4). In rural areas, for children between 0 and 59 months of age an additional year of a mother's schooling leads to -2.3%, or (0.977–1)*100%, change in the odds of being stunted. In urban areas an increase in a mother's years of education does not affect the odds of a child being stunted. The average educational attainment of mothers is more than 5 years greater in urban areas than in rural areas, which suggests that in contexts where the educational attainment is relatively high, additional years of mother's education are not likely to result in further improvements in the nutritional status of children as measured by stunting and HAZ. In contrast, in the poorer rural areas where the average level of maternal education is considerably lower, an increase in mother's education appears to have an impact on the stunting of not only older children but also younger cohorts (0 to 35 months old).

## Conclusions

Estimates of the relationship between a mother's education and child stature with early childhood development are contrasted between two empirical approaches: the conventional approach whereby control variables are selected in an ad-hoc manner and the double machine-learning (DML). The DML approach employs data-driven methods to select controls from a much wider set of variables available in the survey along with statistical learning methods that yield information that would not be available from fitting the model only once using the original sample.

It is hoped that the analysis and comparisons carried out in this paper have made it clear that there is much to be gained by complementing, if not moving away from, the customary approach of controlling for a few selected confounding factors in the regression analysis that is primarily equipped to establish partial correlations and not necessarily causal inferences. The DML estimates are preferred because they are estimates of the causal effect with a more reasonable set of assumptions, they minimize potential omitted variable bias, and they allow for the inclusion of different confounding factors influencing the outcome variable (ECD measure) and the variables for which inferences are desired (child HAZ or stunting) and maternal education, in this paper.

Overall, the analysis based on data from Nigeria confirms that maternal education and the incidence of chronic malnutrition have a significant direct effect on measures of early childhood development. The point estimates based on the ad-hoc specification tend to be larger in absolute value than those based on the DML specification. Frequently, the point estimates based on the ad-hoc specification are very close to the estimates based on the DML specification or fall inside the confidence interval of the DML point estimates. This suggests that in these cases the omitted variable bias is not serious enough to prevent making causal inferences based on the ad-hoc specification. However, there are instances where the omitted variable bias is sufficiently large for the ad hoc specification to yield a statistically significant relationship when in fact the more robust DML specification suggests there is none. The most striking difference between the estimates based on the ad-hoc and the DML approaches concerns the direct and indirect effect of maternal education in urban and rural areas. The ad-hoc estimates suggest that maternal education has a direct effect as well as an indirect effect through the channel of improved child nutrition on the ECD measures of children. The DML approach reveals a more complex picture that highlights the role of context. In rural areas, mother's education affects early childhood development both directly and indirectly through its impact on the nutritional status of both older and younger children. In contrast, in urban areas, where the average level of maternal education is much higher, increases in a mother's education have only a direct effect on child ECD measures but no indirect effect through child nutrition. That is, relying on the estimates from the ad hoc regression model leads to suboptimal policy prescriptions. In urban areas a focus on mother's education does not imply better nutritional outcomes for children whereas in the rural areas focusing on mother's education both improves child nutritional as well as ECD outcomes.

The DML approach offers the opportunity to identify and validate causal relationships with country-specific observational data. Given the constraints faced by randomized control trials to address every possible question, DML provides a practical and feasible approach to reducing threats to internal validity to derive robust inferences and policy design based on observational data. The extent to which the DML estimates are close to the true causal or impact estimates has yet to be established and this can only be ascertained by comparing with causal estimates derived from an RCT. Nevertheless, there is promising evidence that estimates based on the DML represent an improvement over the usual OLS in the sense that they are closer (and

sometimes quite close) to the impact estimates from the preferred instrumental variable method [42].

## Supporting information

**S1 Appendix. The steps involved in the Double Machine Learning (DML) or cross-fit par-tialling-out approach.**
(DOCX)

**S1 Table. Child, mother and household characteristics.**
(DOCX)

**S2 Table. Tables of DML results by zone.**
(DOCX)

## Author Contributions

**Conceptualization:** Emmanuel Skoufias, Katja Vinha.

**Data curation:** Emmanuel Skoufias, Katja Vinha.

**Formal analysis:** Emmanuel Skoufias, Katja Vinha.

**Funding acquisition:** Emmanuel Skoufias.

**Investigation:** Emmanuel Skoufias, Katja Vinha.

**Methodology:** Emmanuel Skoufias, Katja Vinha.

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
