## [Decision Letter · Decision Letter 0]

1 Apr 2021

PONE-D-21-00910

Child Stature, Maternal Education, and Early Childhood Development in Nigeria

PLOS ONE

Dear Dr. Skoufias,

Thank you for submitting your manuscript to PLOS ONE. After careful consideration, we feel that it has merit but does not fully meet PLOS ONE’s publication criteria as it currently stands. Therefore, we invite you to submit a revised version of the manuscript that addresses the points raised during the review process.

We look forward to receiving your revised manuscript.

Kind regards,

Srinivas Goli, Ph.D.

Academic Editor

PLOS ONE

Journal Requirements:

Additional Editor Comments (if provided):

Considering the mixed opinion from the reviewers, I am going with the decision of "Minor revision" before recommending this piece for publication. Reviewers have highlighted some caveats which must be revised before recommending your piece for publication in PLOS One.

Reviewers' comments:

Reviewer's Responses to Questions

**Comments to the Author**

1. Is the manuscript technically sound, and do the data support the conclusions?

Reviewer #1: Yes

Reviewer #2: Partly

Reviewer #3: No

Reviewer #4: Yes

2. Has the statistical analysis been performed appropriately and rigorously? 

Reviewer #1: Yes

Reviewer #2: I Don't Know

Reviewer #3: No

Reviewer #4: Yes

3. Have the authors made all data underlying the findings in their manuscript fully available?

Reviewer #1: Yes

Reviewer #2: Yes

Reviewer #3: Yes

Reviewer #4: Yes

4. Is the manuscript presented in an intelligible fashion and written in standard English?

Reviewer #1: Yes

Reviewer #2: Yes

Reviewer #3: No

Reviewer #4: Yes

5. Review Comments to the Author

Reviewer #1: This paper brings out an important approach to looking at a common child health question and shows clearly the value of employing DML as opposed to the conventional method to bring out the relationship between maternal education, ECD and child stature in a LMIC.

Reviewer #2: The paper addresses the issue of efficacy of the Randomized controlled trials[ called ad-hoc approach in the paper] vis a vis a direct machine learning[ DML] approach where using artificial intelligence, controls can be assigned across different variables

The results of the two approaches are interesting, and sometimes conflicting. One example being that in both rural and urban areas, the DML approach suggests that mother education has no impact on a child being on track in learning, whereas the ad-hoc approach suggests that it does. Similarly, mother’s education impact on a child’s chronic malnutrition status measured by whether the child is stunted differs in both methods in rural and urban areas.

There are two basic issues here. One is from the policy perspective- in the face of conflicting results, which one should the policy maker rely on? Often the data quality and its limitations in all their granularity are not know at the top levels of decision making. The interventions would differ widely with two starkly different pieces of evidence based on the same set of people. This contradiction needs to be addressed by the authors. The paper may delve deeper into the faithful adherence to reality and the offering for the policy makers for both approaches and have a clear set of pros and cons for each, which can be taken into account for decision making.

Secondly, RCTs are designed to eliminate biases in allocation of samples to control and intervention groups. At the same time, AI is not free of biases and in fact may “internalize” them based on the data sets. This is an ethical limitation that may skew the interpretation. The authors may address this issue as well, looking into the bias replication by AI.

Overall, with these caveats , it is an interesting and novel study, and can contribute to furthering and honing the filed of data analysis and interpretation.

Reviewer #3: Dear Author,

Thank you for your interest in this journal. The manuscript in it's current form did not meet the criteria for publication in this journal as the guidelines for submission were not adhered to.

Reviewer #4: the language that been used is to technical, should try to simplified so it can reach out more wider reader

definition of stunted and stunting should be described more clearly

does any other social factor already measures (like infection and sanitation)

6. PLOS authors have the option to publish the peer review history of their article (what does this mean?). If published, this will include your full peer review and any attached files.

Reviewer #1: No

Reviewer #2: No

Reviewer #3: No

Reviewer #4: No

---

## [Author Response · Author response to Decision Letter 0]

27 Jul 2021

In revising our paper we have taken into consideration all the comments received from the reviewers including, adhering to the submission standards, of the journal (reviewer 3), the use of a more simplified language so we can reach out more wider audience (Reviewer 2), and a more clear definition of stunting or being stunted (reviewer 4). We have not addressed some of the points of reviewer 2 as we think they are based on a misunderstanding of what this paper does. For example, reviewer 2 prefaces his/her comments with the statement: ”The paper addresses the issue of efficacy of the Randomized controlled trials[ called ad-hoc approach in the paper] vis a vis a direct machine learning[ DML] approach where using artificial intelligence, controls can be assigned across different variables” 

Our paper does not address the issue of efficacy of RCTs. It simply states that RCTs are not possible to implement to address many policy relevant questions such as the one in the paper: “Does mother’s education reduced child stunting and is child stunting associated with lower ECD?” The ad-hoc approach we refer to is the ad-hoc way of choosing controls in a regression-based approach intended to examine the relation (ideally the casual relation) between stunting, mother’s education and ECD.

---

## [Decision Letter · Decision Letter 1]

22 Nov 2021

Child Stature, Maternal Education, and Early Childhood Development in Nigeria

PONE-D-21-00910R1

Dear Dr. Skoufias,

We’re pleased to inform you that your manuscript has been judged scientifically suitable for publication and will be formally accepted for publication once it meets all outstanding technical requirements.

Kind regards,

Srinivas Goli, Ph.D.

Academic Editor

PLOS ONE

Additional Editor Comments (optional):

Considering my own reading and reviewers opinion, I am recommending this paper for publication in PLOS One.

Reviewers' comments:

Reviewer's Responses to Questions

**Comments to the Author**

1. If the authors have adequately addressed your comments raised in a previous round of review and you feel that this manuscript is now acceptable for publication, you may indicate that here to bypass the “Comments to the Author” section, enter your conflict of interest statement in the “Confidential to Editor” section, and submit your "Accept" recommendation.

Reviewer #4: (No Response)

2. Is the manuscript technically sound, and do the data support the conclusions?

Reviewer #4: Yes

3. Has the statistical analysis been performed appropriately and rigorously? 

Reviewer #4: Yes

4. Have the authors made all data underlying the findings in their manuscript fully available?

Reviewer #4: Yes

5. Is the manuscript presented in an intelligible fashion and written in standard English?

Reviewer #4: Yes

6. Review Comments to the Author

Reviewer #4: the input that been given already been corrected.. but the importance of this research for daily practice could be added in the title so it will attract more reader

7. PLOS authors have the option to publish the peer review history of their article (what does this mean?). If published, this will include your full peer review and any attached files.

Reviewer #4: No

---

## [Editor Report · Acceptance letter]

6 Dec 2021

PONE-D-21-00910R1 

Child Stature, Maternal Education, and Early Childhood Development in Nigeria 

Dear Dr. Skoufias:

I'm pleased to inform you that your manuscript has been deemed suitable for publication in PLOS ONE. Congratulations! Your manuscript is now with our production department. 

Kind regards, 

on behalf of

Dr. Srinivas Goli 

Academic Editor

PLOS ONE